# IL-27 Signaling Promotes Th1 Response by Downregulating IL-10 Production in DCs during Chlamydial Respiratory Infection

**DOI:** 10.3390/microorganisms11030604

**Published:** 2023-02-27

**Authors:** Jiajia Zeng, Shuaini Yang, Yuqing Tuo, Xiaoyu Zha, Ruoyuan Sun, Tingsha Lu, Hong Zhang, Lu Tan, Sai Qiao, Hong Bai

**Affiliations:** Key Laboratory of Immune Microenvironment and Disease (Ministry of Education), Department of Immunology, School of Basic Medical Sciences, Tianjin Medical University, Tianjin 300070, China

**Keywords:** chlamydial infection, IL-27, dendritic cells, maturation, Th1 response, IL-10

## Abstract

*Chlamydia trachomatis* usually causes mucosal infections, bringing considerable morbidity and socioeconomic burden worldwide. We previously revealed that IL-27/IL-27R mediates protection against chlamydial invasion by promoting a protective Th1 response and suppressing neutrophilic inflammation. Here, we used the mouse model of *Chlamydia muridarum* (*C. muridarum*) respiratory infections to further investigate the impact of IL-27 signaling in the DCs-regulated immune response, since an elevated IL-27/IL-27R expression in DCs was identified following chlamydial infection. An adoptive transfer of *Chlamydia muridarum*-stimulated DCs to wild-type mice approach was subsequently used, and the donor-DCs-promoted resistance with a higher Th1 response against chlamydial infection was attenuated when DCs lacking IL-27R were used as donor cells. Flow cytometry analysis revealed the suppression of IL-27 signaling on DCs phenotypic maturation. A further functional maturation analysis of DCs revealed that IL-27 signaling restricted the protein and mRNA expression of IL-10 from DCs following infection. Thus, these findings suggest that IL-27 signaling could support the Th1 response via inhibiting IL-10 production in DCs, thus mediating the protective host defense against chlamydial respiratory infection.

## 1. Introduction

As an obligate intracellular pathogen, *Chlamydia trachomatis* (*C. trachomatis*) frequently causes mucosal infection through the eye, genital tract, and respiratory system, resulting in trachoma, urogenital tract infection, and neonatal pneumonia [1,2,3]. Moreover, *C. trachomatis* infection can induce immunopathological damage by delayed-type hypersensitivities, such as lymphogranuloma of sexually transmitted diseases [4]. The host typically displays a weak defense with persistent and asymptomatic infection; therefore, recognizing and treating infected hosts is key to preventing irreparable tissue damage and controlling chlamydial illnesses. *C. trachomatis* mouse pneumonitis biovar (*C. muridarum*) is a natural pathogen of mouse pneumonia, and the mouse model of *C. muridarum* infection is widely used for chlamydial infection in vivo [5,6]. The *C. muridarum*-induced Th1 response exerts a protective effect through the production of interferon-gamma (IFN-γ), which was reported to enhance the phagocytic capacity of macrophages and directly inhibit the growth of chlamydia by promoting the synthesis of inducible nitric oxide synthase (iNOS) and indoleamine 2, 3-dioxygenase (IDO) [7,8]. By modifying the DC function, IL-17/Th17 was found to boost the Th1 response in our earlier research [3], but excessive IL-17 could cause immunopathological effects by eliciting neutrophilic inflammation [9].

Dendritic cells (DCs), as innate immune cells and professional antigen-presenting cells, are crucial for resisting pathogen invasion and initiating and regulating adaptive immunity [10,11]. As reported, DCs actively participate in the regulation of the chlamydia-induced host response. Chlamydia might remain alive and infectious in DCs [12]. Chlamydia inclusion bodies were demonstrated to stimulate the maturation and generation of the IL-12 and TNF of DCs; adoptive transferring DCs enhanced resistance against chlamydia [13]. We previously showed that IL-17 neutralization inhibits the chlamydia-specific Th1 response and decreases DC’s phenotypic maturation and IL-12 secretion. Furthermore, adoptively transferring DCs isolated from IL-17-neutralized mice induces a diminished protective effect on recipient mice, suggesting that the IL-17/Th17’s promoting effect on the Th1 response is achieved by regulating the DC activity and function [3]. Our partner Dr. Yang further compared the contributions of DC subsets in chlamydia defense. Following the adoption of CD8α^+^ splenic DCs, Th1-related cytokines were considerably higher than those following the adoption of CD8α^−^ DCs [14]. Similarly, mice-receiving CD103^+^ LDCs exhibited better protection than CD11b^hi^ LDCs recipients, correlated with more robust Th1/Th17 responses [15]. The above documents indicate that DCs pose significant regulatory effects in *C. trachomatis* infection.

Known as the “Immunological Playmakers”, the Interleukin (IL)-12 cytokines family is the only family of heterodimeric cytokines, which endows them with several unique connections and functional interactions [16]. IL-27, a member of the IL-12 family, is composed of a unique IL-12p35-like protein, IL-27p28, and a protein related to IL-12p40 encoded by the Epstein–Barr-virus-induced gene 3 (EBI3) [17,18]. The dominant cellular sources of IL-27 are reported to be DCs and macrophages, although other myeloid cell populations such as monocytes, microglia, and epithelial cells also express IL-27 [19]. IL-27 signals through a heterodimeric surface receptor consist of glycoprotein 130 (GP130) and IL-27Rα (also known as WSX-1) [20]. Being implicated in a variety of infectious diseases, IL-27 is recognized as both immunosuppressive and immunostimulatory in innate and adaptive immunity. In research on post-influenza pneumococcal pneumonia, IL-27 sensitized mice to secondary pneumococcal infection, attributable to the inhibition of IL-17A production in γδ T cells in a STAT1-dependent manner [21]. In *Clostridium difficile* colitis infection, recombinant IL-27 administration caused an increased IFN-γ with a decreased mortality in mice [22]. In secondary *Staphylococcus aureus* pneumonia following influenza infection, IL-27 caused an enhanced susceptibility by inducing IL-10 and suppressing IL-17 [23]. In *C. muridarum*-induced respiratory infection, our previous study found that IL-27/IL-27R protects the host by promoting the Th1 response and suppressing excessive IL-17-induced neutrophilic inflammation [9]. As DCs are involved in chlamydial infection and its regulation on T cell responses, we hypothesized that DCs participate in the IL-27/IL-27R’s protective effect during chlamydial respiratory infection.

Using the *C. muridarum-*infected mouse model, we were able to identify how IL-27/IL-27R affected DCs maturation as well as the associated immunological effect and mechanism. Though IL-27 signaling inhibited the accumulation and phenotypic maturation of DCs, the adoptive transfer experiment revealed that WSX-1^−/−^-DC recipients experienced a worsened disease with weaker Th1 levels. Further exploration found an inhibitory effect of IL-27/IL-27R on IL-10 production by DCs. Our findings thus indicate that IL-27 signaling could promote the protective Th1 response by regulating the cytokine production by DCs in *C. muridarum* infection, which may provide novel insights into the treatment of intracellular bacterial infection.

## 2. Materials and Methods

### 2.1. Animals

Wild-type (WT) female C57BL/6 mice (6–8 weeks old, 18–20 g) were purchased from Huafukang Biotechnology Co., Ltd. (Beijing, China). Female IL-27Rα-deficient (WSX-1^−/−^) mice on the C57BL/6 background were granted by Professor Yin Zhinan (Nankai University, China). All mice were fed in specific pathogen-free (SPF) circumstances at Tianjin Medical University (Tianjin, China). All procedures involving animals were approved by the Animal Ethical and Welfare Committee (AEWC) of Tianjin Medical University (number of animal permit: SYXK: 2016-0012, approval date: 7 March 2018).

### 2.2. Bacterial Strains, Chlamydial Infection and Administration of rIL-27

*Chlamydia muridarum* (*C. muridarum*) strains were first gifted from Dr. Xi Yang (the University of Manitoba, Canada) and cultured, and then expanded in our laboratory as previously described [15]. To induce respiratory infection, WT and WSX-1^−/−^ mice were anesthetized with isoflurane, intranasally (i.n.) inoculated with 1 × 10^3^ inclusion-forming units (IFUs) of *C. muridarum* in 40 μL sucrose–phosphate–glutamic acid buffer (SPG), with the uninfected mice (0 d) inoculated 40 μL SPG buffer as a control. For administration of recombinant murine IL-27 (rIL-27) (BioLegend, San Diego, CA, USA), WT mice were anesthetized, i.n. inoculated with 0.2 μg rIL-27 in 30 μL PBS the day before and days 0, 2, 4, and 6 after *C. muridarum* infection or 30 μL aseptic protein-free PBS on the same schedules as the control. Following infection, body weight was recorded daily, and mice were sacrificed at the indicated time points.

### 2.3. Lung, Spleen, and Lymph Nodes (LNs) Single-Cell Preparation

*C. muridarum*-infected lung tissues were minced and digested with 2 mg/mL collagenase XI (Sigma-Aldrich, St. Louis, MO, USA) in PRMI-1640 for 55 min at 37 °C. Tissue fibers and erythrocytes were successively removed by 35% Percoll (GE Healthcare, Chicago, London, UK) and ACK Lysis buffer (Tris-NH_4_Cl). *C. muridarum*-infected spleens and LNs were directly ground and filtered through 70 μm cell strainers, and erythrocytes were lysed using an ACK lysing buffer. These single cells were resuspended in the complete RPMI-1640 medium (RPMI-1640 supplemented with 10% heat-inactivated FBS, 0.05 mmol/L 2-mercaptoethanol, 100 U/mL penicillin, and 0.1 mg/mL streptomycin), stained with trypan blue, and then counted under light microscopy for further analysis.

### 2.4. Antibodies and Flow Cytometry

All of the antibodies were indicated in Appendix A. Single-cell suspensions were resuspended in FACS buffer (PBS with 2% FBS) and incubated with Fc receptor block Abs (anti-CD16/CD32 monoclonal Abs; eBioscience, San Diego, CA, USA) at 4 °C for 20 min in the dark. For cell surface staining, cells were stained with conjugated antibodies specific for cell surface markers at 4 °C for 30 min in the dark. For intracellular cytokine staining, cells were firstly stimulated with cocktails including PMA (50 ng/mL, Solarbio, Beijing, China), Ionomycin (1 μg/mL, MCE, Monmouth Junction, NJ, USA), and brefeldin A (10 μg/mL, BioLegend) at 37 °C for 5–6 h. Following a blocking step similar to cell surface staining, cells were stained for surface antigens and fixed with the Fixation buffer (Biolegend). Fixed cells were washed and permeabilized with 1 × Intracellular Staining Perm Wash Buffer (Biolegend) followed by incubation with anti-IFN-γ, anti-IL-4, or anti-IL-17A mAb for 30 min at room temperature. Finally, cells were suspended with FACS buffer, detected using a FACSCanto II flow cytometer (BD Biosciences, Franklin, NJ, USA), and analyzed by the FlowJo V10 software.

### 2.5. Generation and Stimulation of Bone-Marrow-Derived DCs (BMDCs)

Bone marrows were flushed from the femurs and tibias of naïve C57BL/6 mice, clusters were dispersed by vigorous pipetting, and erythrocytes were lysed by ACK buffer. Cells were washed and cultured at 1–2 × 10^6^ /mL in the complete RPMI-1640 medium supplemented with 20 ng/mL murine GM-CSF (R&D Systems, Madrid, Spain) and 10 ng/mL murine IL-4 (PeproTech, Cranbury, NJ, USA) in 6-well cell culture plates. A total of 20 ng/mL rIL-27 was added or not added to the BMDC culture media to compare the impact of IL-27/IL-27R. On day 3, the entire medium was discarded and replaced with fresh medium. On day 5, half of the culture supernatant was removed and replaced with 2 mL fresh culture medium. On day 7, non-adherent cells were collected and transferred to 10 mm cell culture dishes. On day 9, suspending cells were harvested and the purity of CD11c^+^ BMDCs was >80% as determined by flow cytometry. For maturation, BMDCs were stimulated with 1 μg/mL LPS (Sigma-Aldrich) for 48 h. With the presence of LPS, 1 × 10^6^ IFUs of *C. muridarum* was added to the medium. Finally, cells were harvested for flow cytometry and PCR, and the supernatants were harvested for ELISA.

### 2.6. Splenic DCs Isolation and Adoptive Transfer

Mice were i.n. infected with *C. muridarum* and spleens were harvested on day 7 following infection. The spleens were minced and digested with 1 mg/mL of collagenase D (Sigma-Aldrich) in RPMI-1640 for 25 min at 37 °C. The cell suspensions were filtered through 70 μm cell strainers, and erythrocytes were lysed by ACK lysing buffer. After the lysis, the cell suspension was incubated with CD11c microbeads (Miltenyi Biotec, Auburn, CA, USA) for 15 min at 4 °C and resuspended in MACS buffer (PBS with 2% FBS and 2 mM EDTA). After incubation, the splenic cells passed through magnetic columns for positive selection of CD11c^+^ cells. The purity of the sorted splenic DCs was more than 90% as detected by flow cytometry. Freshly isolated CD11c^+^ spleen cells were adoptively transferred to syngeneic naïve C57BL/6 mice through tail vein injection at 1 × 10^6^ DCs/mouse in 200 μL PBS, with mice injected 200 μL PBS taken as control. Two hours later, the recipient mice were intranasally inoculated with 40 μL SPG containing 1 × 10^3^ IFUs *C. muridarum*. Following infection, body weight was recorded daily, and recipient mice were sacrificed on day 14 post-infection (p.i.).

### 2.7. Co-Culture of Splenic DCs and CD4^+^ T Cells

CD11c^+^ DCs were isolated from the spleen of *C. muridarum*-immunized mice on day 7 p.i. or naïve C57BL/6 mice using magnetic beads and MACS columns through positive selection (Miltenyi Biotec, Auburn, CA, USA) as described above. CD4^+^ T cells were isolated from the spleen of *C. muridarum*-immunized mice on day 7 p.i. or naïve C57BL/6 mice using magnetic beads and MACS columns through negative selection (Miltenyi Biotec, Auburn, CA, USA) as described previously [24]. Briefly, spleen single-cell suspensions were mixed with Biotin-Antibody Cocktail, then incubated with Anti-Biotin MicroBeads. Labeled cells passed through magnetic columns for negative selection of CD3^+^ CD4^+^ T cells. As shown by flow cytometric analysis, the purity of the CD4^+^ T cells was more than 96%. CD4^+^ T cells were co-cultured with CD11c^+^ DCs (DC/T cell ratio, 1:5; CD11c^+^ DCs: 1 × 10^5^, CD4^+^ T cells: 5 × 10^5^) in 96-well plates in the presence or absence of UV-sterilized *C. muridarum* (UV-Cm) for 48h. Cell supernatants were collected for analyzing IFN-γ, IL-4, and IL-17 production by ELISA.

### 2.8. Pulmonary Chlamydial Loads

Chlamydia IFUs were detected to determine the growth of *C. muridarum* in infected lungs as described previously [9,25]. Briefly, the lung tissues were homogenized aseptically in SPG buffer, acquired lung homogenates were centrifuged at 3000 rpm for 30 min, and diluted supernatants were added into a confluent monolayer of HeLa cells and incubated at 37 °C for 2 h. After being cultured in complete RPMI-1640 medium for 24 h, media were removed and infected cells were fixed with methanol for 10 min. The cells were then stained with anti-Chlamydia LPS antibody (Invitrogen, Carlsbad, CA, USA) and HRP-conjugated goat anti-mouse IgG secondary Abs (Solarbio) and developed with the substrate (4-chloro-1-naphthol; Solarbio). The inclusion bodies were counted under the microscope [100X] to calculate IFUs per sample.

### 2.9. Histology Analysis and Semi-Quantitative Pathological Scoring

For histopathological analyses, *C. muridarum*-infected lung tissues were fixed in 10% formalin and routinely embedded in paraffin, sectioned (5 μm), and stained with hematoxylin and eosin (H&E). The pathological changes were evaluated by the semi-quantitative histology score in a blinded manner as described previously [9] and introduced in Appendix A.

### 2.10. ELISA

The spleen DCs isolated from *C. muridarum*-infected WT and WSX-1^−/−^ mice on day 7 p.i., T cells co-cultured with DC, and BMDCs under different treatments were cultured to test *C. muridarum*-driven cytokine production. Briefly, these cell suspensions were cultured at a concentration of 5 × 10^5^ splenic DCs/well, 5 × 10^5^ CD4^+^ T cells with 1 × 10^5^ DCs/well, and 1 × 10^6^ BMDCs/well, respectively. After 48/72 h culture, the supernatants were collected for cytokine detection. IFN-γ, IL-4, IL-17, IL-12p40, IL-10, and IL-6 productions were measured by ELISA according to the manufacturer’s instructions (Invitrogen). The OD values were read at 450 nm on the Epoch microplate reader (BioTeK, Winooski, VT, USA).

### 2.11. RNA Extraction and Quantitative Real-Time PCR (qPCR)

Total RNA of sorted spleen DCs and BMDCs was extracted by TRIzol reagent (Invitrogen) and reverse transcription was performed using the cDNA Synthesis SuperMix (TransGen Biotech, Beijing, China) according to the manufacturer’s instructions. qPCR was further performed using RealStar Fast SYBR qPCR Mix (GenStar, Beijing, China) on Light Cycler 96 (Roche, Basel, Switzerland). The mRNA expression of target genes was presented as the “fold change” relative to that of control samples, with fold changes calculated by the 2^−ΔΔCt^ method using the mouse β-actin gene as an endogenous control. The PCR primer sequences used are shown in Appendix A.

### 2.12. Statistical Analysis

Data are represented as means ± SD and were assessed with GraphPad Prism 9. The significance of differences in two different groups was analyzed by unpaired Student’s *t*-test or two-way ANOVA followed by Šidák’s multiple comparisons test; differences between multiple groups were analyzed by one-way ANOVA followed by Dunnett’s multiple comparisons test. *p* values < 0.05 were considered significant (* or # *p* < 0.05, ** or ## *p* < 0.01, *** or ### *p* < 0.001, **** or #### *p* < 0.0001).

## 3. Results

### 3.1. C. muridarum Respiratory Infection Induces IL-27/IL-27R Expression of DC

We recently discovered that *C. muridarum* infection aggravated the disease in WSX-1^−/−^ mice, with a reduced Th1 response and IL-17-induced neutrophilic inflammation [9]. Given that DCs boosted the Th1 and Th17 responses against *C. muridarum* invasion, it was conceivable that DCs contributed to the protective effects of IL-27 signaling. Using flow cytometry, we identified pulmonary and splenic DCs as CD45^+^ CD11c^+^ MHCII^+^ cells (Appendix A). The expressions of WSX-1 on pulmonary and splenic DCs gradually increased from day 3 post-infection (p.i.) and peaked on day 7 p.i. (Figure 1A,B). After sorting the splenic DCs, qPCR was conducted to detect the mRNA expression of IL-27 (composed of p28 and EBI3) and IL-27R (composed of WSX-1 and GP130) (Figure 1C,D). The mRNA level of p28 was higher on day 3 p.i. and WSX-1 increased on day 7 p.i. compared with uninfected controls. To further confirm these findings, we cultured BMDCs and identified an increased WSX-1 expression after the *C. muridarum* challenge, with the LPS treatment taken as the positive control (Figure 1E). These data indicated an increased expression of IL-27 and IL-27R on DCs following infection, suggesting that DCs may participate in the IL-27/IL-27R-modulated host defense against *C. muridarum* infection.

### 3.2. Adoptive Transfer of WSX-1^−/−^ DC Reduces Protection against C. muridarum Respiratory Infection

To assess the impact of IL-27/IL-27R-modulated DCs, we compared the immunopathological consequences of recipient mice adoptively transferred with splenic DCs derived from WT (WT-DC) and WSX-1^−/−^ (WSX-1^−/−^-DC) mice. As shown in Figure 2A,B, the WT-DCs and WSX-1^−/−^-DCs were sorted with purity validated (Figure 2B) and adoptively transferred into recipients, with mice receiving PBS used as control. As reported, the adoptive transfer of DCs results in a protective effect, as WT-DC recipients developed fewer body weight losses and decreased IFUs (indicating chlamydial burden) than the PBS group (Figure 2C,D). Compared with WT-DC recipients, WSX-1 deficiency caused greater body weight losses and higher IFUs, suggesting the negative regulation of WSX-1^−/−^-DCs on the host defense. In agreement with the above conditions, *C. muridarum* respiratory infection induced a pulmonary inflammatory pathology, while WSX-1^−/−^-DC recipients displayed graver inflammatory grades with more massive inflammatory cell infiltration than WT-DC recipients (Figure 2E,F). These results show that the adoptive transfer of WSX-1^−/−^-DCs failed to protect the recipients from *C. muridarum* infection more effectively than WT-DCs.

### 3.3. Adoptive Transfer of WSX-1^−/−^ DC Reduces Th1 Responses following C. muridarum Respiratory Infection

The Th1 and Th17 responses were reported to be protective against *C. muridarum* infection, whereas the Th2 response causes pathological effects [3,26]. Importantly, we reported that IL-27/IL-27R suppresses neutrophilic inflammation [9]. Thus, Th responses and neutrophil levels of recipient mice were detected to clarify the regulating target of WSX-1^−/−^-DC. Consistent with disease conditions, the percentage of lung Th1 cells was higher in WSX-1^−/−^-DC recipients than in PBS recipients but lower in WT-DC recipients (Figure 3A). Similar changes were observed in the spleen. Figure 3B–D illustrate that WSX-1 deficiency elicited an increased neutrophil infiltration in the lung, whereas the Th2 and Th17 responses were comparable between the WT-DC and WSX-1^−/−^-DC recipients.

To further confirm the impact of IL-27/IL-27R-modulated DCs on Th responses, splenic DCs and CD4^+^ T cells were sorted and co-cultured. As indicated in Figure 3E, in the absence of UV-Cm, neither the naïve T cells nor Cm-T cells alone induced IFN-γ secretion. Under UV-Cm stimulation, there were no differences in the amount of IFN-γ when naïve CD4^+^ T cells and DCs were co-cultured. When *C. muridarum*-immunized T cells were co-cultured with DCs, infected WSX-1^−/−^-DCs secreted fewer IFN-γ than infected WT-DCs did. The IL-4 and IL-17 production in the co-culture system does not differ significantly from the flow cytometry analysis in Figure 3B,C. All of these results suggested that DCs derived from WSX-1^−/−^ mice could weaken the host immunity against *C. muridarum* respiratory infection by suppressing the Th1 response.

### 3.4. Deficiency of WSX-1 Promotes the Accumulation and Phenotypic Maturation of DCs following Chlamydial Respiratory Infection

Next, we compared DCs infiltration and maturation between *C. muridarum*-infected WT and WSX-1^−/−^ mice for clarification of the mechanism by which WSX-1^−/−^-DCs inhibit the Th1 response. Following *C. muridarum* invasion, DCs accumulated to the lung and spleen, while WSX-1^−/−^ mice exhibited more pulmonary DCs on days 3 and 7 p.i. and increased splenic DCs in both the percentage and absolute number on day 7 p.i. (Figure 4A,B).

DC maturation is determined by changes at the phenotypic and functional levels [27,28,29]. When DCs increase the expression of costimulatory molecules CD40, CD80, and CD86, as well as the MHC class II molecule, phenotypic maturation occurs [30]. DCs at an elevated functional level typically secrete cytokines, including inflammatory cytokines and immunosuppressive cytokines [31]. When WSX-1^−/−^ mice were compared to WT mice on day 7 p.i., flow cytometry analysis showed a higher percentage and mean fluorescence intensity (MFI) of CD80 on pulmonary DCs in WSX-1^−/−^ mice (Figure 4C,D). Similarly, on day 7 p.i., splenic DCs from WSX-1^−/−^ mice displayed increased percentages and MFIs of CD40, CD80, and CD86 (Figure 4E,F). These findings indicated that the accumulation and phenotypic maturation of DCs during *C. muridarum* infection may be constrained by IL-27 signaling.

### 3.5. rIL-27 Treatment Suppresses the Accumulation and Phenotypic Maturation of DC following Chlamydial Respiratory Infection

To further confirm the impact of IL-27 signaling on DCs accumulation and phenotypic maturation, recombinant murine IL-27 (rIL-27) or an equal volume of PBS was given to naïve C57BL/6 mice that were subsequently challenged with *C. muridarum*. On day 7 following infection, the rIL-27-treated mice developed a lower percentage of DCs than control mice in the spleen (Figure 5A). On day 14 following infection, a reduced CD40 expression on pulmonary DCs was observed in rIL-27-treated mice when compared to PBS-treated mice (Figure 5B). Similar patterns of costimulatory molecules, such as CD40, CD80, and CD86 on splenic DCs and CD40 on BMDCs, were seen in the groups treated with rIL-27 (Figure 5C,D). Collectively, these data further support the role of IL-27 signaling in limiting DCs accumulation and phenotypic maturation during *C. muridarum* infection.

### 3.6. IL-27 Signaling Inhibits IL-10 Production in DCs following C. muridarum Respiratory Infection

DCs-derived cytokines are essential for modulating the adaptive immune response. Next, we investigated the DC cytokines profile to identify its functional maturation [31,32]. Among them, DCs-derived IL-12 is essential for the generation of the Th1 cell response, whereas IL-10 and IL-6 can inhibit it [33]. The uninfected and infected spleen DCs were sorted from the wild-type and WSX-1^−/−^ mice for qPCR and ELISA tests. While mRNA expressions of IL-12p40 and IL-12p35 are comparable, the qPCR results demonstrated an increased IL-10 and IL-6 expression in WSX-1^−/−^-DCs compared to WT-DCs on day 7 p.i. (Figure 6A). Further, in the ELISA analysis, the amount of IL-10 in the culture supernatant of WSX-1^−/−^-DCs was significantly higher than WT-DCs (Figure 6B). To further confirm the cytokines production, BMDCs were treated or not treated with rIL-27, and the qPCR results show that *C. muridarum* infection induced an increased IL-12p40, IL-12p35, IL-10, and IL-6 mRNA expression, and that rIL-27 treatment inhibited the level of IL-10 and IL-6 (Figure 6C). ELISA further supported the inhibition of rIL-27 to IL-10 and IL-6 production on the protein level (Figure 6D). Together, these data demonstrate a mechanistic connection between IL-27 signaling and DCs-secreted IL-10, which suppress the protective Th1 response during *C. muridarum* lung infection.

## 4. Discussion

IL-27 signaling was recently demonstrated to mediate protective immunity against chlamydial infection [9]. Here, we further determined that *C. muridarum* infection induces the WSX-1 expression on DCs, and the adoptive transfer of WSX-1^−/−^ mice-derived DCs failed to mediate effective protection, attributable to a diminished Th1 response. Further investigation revealed that WSX-1^−/−^-DCs increased the IL-10 production compared to WT-DCs, and that the rIL-27 administration reduced IL-10 secretion, suggesting the inhibition of IL-27 on the Th1 response through raising IL-10 production from DCs. This study firstly reveals IL-27/IL-27R’s encouraging role in the Th1 response by regulating DCs cytokines during chlamydial infection.

The WSX-1^−/−^ mice displayed a lower Th1 response and a higher Th17 response with excessive neutrophilic inflammation [9], so we compared the Th1/2/17 response and neutrophil infiltration between WSX-1^−/−^-DC and WT-DC recipient mice. Consistently with a reduced protection, the WSX-1^−/−^ DC recipient conferred a lower Th1 response (Figure 3A) and increased neutrophil infiltration, as well as no difference with the PBS recipient (Figure 3D) and comparable Th2 and Th17 levels. These data suggest that though IL-27 signaling has a regulatory role on the Th1 and Th17 response, IL-27/IL-27R-modulated DCs primarily affect the Th1 response. These data potently indicate the link between DC and T-cell-mediated responses, and, more importantly, also explain why we focus on measuring Th1-related cytokines in subsequent experiments.

Identified as an anti-inflammatory factor and a Th2 cytokine, IL-10 was engaged in inflammatory responses and controlling the Th1 response [34,35]. During acute *Clostridioides difficile* infection, IL-10^−/−^ mice exhibited an elevated IL-22 response and decreased mortality [36]. Sarah E. Clark et al. reported that, in *Streptococcus pneumoniae (S. pneumoniae)* infection, NK-dependent IL-10 restricts myeloid cell recruitment, causing an increased bacterial growth and exacerbated infection [37]. In *Mycobacterium tuberculosis* (*M*. *tuberculosis*) infection, early IL-10 overexpression reduces the T cells’ capacity for parenchymal migration and interacting with infected phagocytic cells, impeding the limitation of *M*. *tuberculosis* growth [38]. In *Schistosomiasis mansoni* infection, a lack of IL-10 caused dysregulated Th1 responses, resulting in aggravated granulomatous inflammation and an increased mortality in the acute phase [39]. In *C. trachomatis* respiratory infection, IL-10 was reported to inhibit the priming and expansion of Th1 responses and contribute to the fibrotic reaction following infection [40]. In this study, in line with the weaker Th1 response of WSX-1^−/−^-DC recipients in the adoptive transfer experiments, IL-10 expression and secretion in WSX-1^−/−^ splenic DCs were significantly higher than in WT mice, suggesting that IL-27/IL-27R may mediate protective Th1 immunity by suppressing the IL-10 production of DCs in *C. muridarum* respiratory infection.

During *C. muridarum* respiratory infection, previous studies showed that Th1-promoting DCs display a more mature phenotype [3,41]; however, this study prompted more phenotypically mature WSX-1^−/−^-DCs to pose a reduced potency to trigger a powerful Th1 response. Indeed, one of the crucial features of DC biology is its maturation. The immature DCs recognize and capture antigens whereas the mature ones are ineffective; in contrast, the mature DCs are potent stimulators for T-cell-mediated responses, making DCs key regulators of both protective immune responses and tolerance to autoantigens [42,43]. As detailed in the *Results*, the phenotypic maturation is not identical to functional maturation, and the capacity of DCs to produce cytokines is a critical step, as DCs participate in regulating adaptive immunity through producing IL-12, IL-10, and IL-6 [32,44,45]. Combined with the consistency between the higher IL-10 level and weaker Th1 response in the WSX-1^−/−^ mice, we tend to draw the conclusion that, though impacting the mature phenotype, IL-27/IL-27R primarily regulates the Th1 response through modulating the DC cytokines profile in chlamydial infection.

Two major subsets of murine DCs, cDC1 (characterized as CD8α^+^ splenic DCs or CD103^+^ CD11b^−^ pulmonary DCs) and cDC2 (characterized as CD8α^−^ splenic DCs or CD103^−^ CD11b^+^ pulmonary DCs), reported posing distinct potentials for inducing protective immunity against *C. muridarum* respiratory infection [14,15], were detected. As shown in Appendix A, comparable infiltrations of CD8α^+^ DCs and CD8α^−^ DCs were identified in the spleens between infected WT and WSX-1^−/−^ mice. CD8α^+^ DCs and CD8α^−^ DCs from WSX-1^−/−^ mice both express higher CD40, CD80, and CD86 than WT mice, indicating that both subsets have a more mature phenotype. These data might imply that WSX-1 depletion cannot induce a significant difference in the infiltration or phenotypic maturation of splenic DC subsets. Further research is needed for the infiltration and maturation of pulmonary DC subsets, the impacts of IL-27/IL-27R on the functional maturation of DC subsets, and, more significantly, the DC-subsets-driven immune regulations to improve this work.

## 5. Conclusions

Collectively, we demonstrate that IL-27 signaling promotes the protective Th1 response by regulating cytokine production from DCs in chlamydial infection. Based on the IL-27/IL-27R-mediated protection by regulating the Th response proved in the recent work, this study further clarified the potential mechanism, contributing to the in-depth understanding of the pathogenesis of chlamydial infection, which will provide new insights into an effective immunotherapy for controlling bacterial respiratory infections.

## Figures and Tables

**Figure 1 microorganisms-11-00604-f001:**
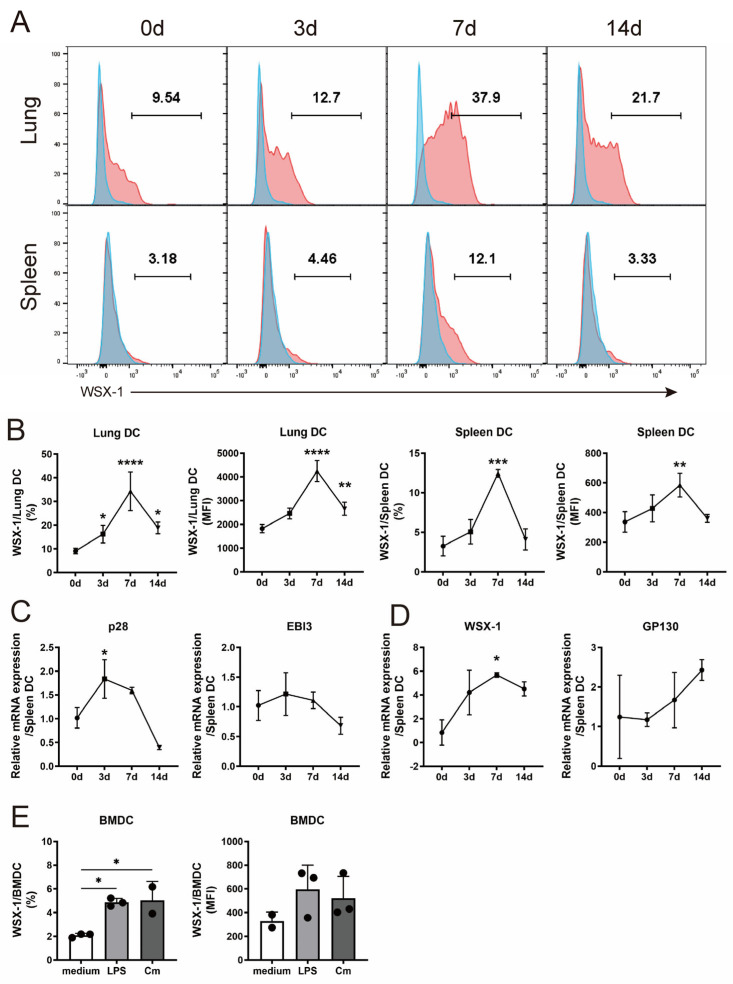
The expression of IL-27/IL-27R in DCs following *Chlamydia muridarum* (*C. muridarum* or Cm) respiratory infection. The wild-type (WT) mice were intranasally infected with 1 × 10^3^ inclusion-forming units (IFUs) *C. muridarum* and euthanized at days 0, 3, 7, and 14 post-infection (p.i.). (**A**,**B**) Lung and spleen single cells were prepared, and the WSX-1 expressions (red) with fluorescence minus one (FMO) control (blue) were analyzed by flow cytometry based on gated DCs as described in Appendix A (**A**). The percentages and mean fluorescence intensity (MFI) of WSX-1 positive cells were indicated (**B**). (**C**,**D**) Spleen CD11c^+^ cells were purified using MACS CD11c^+^ cell isolation column, total RNA was prepared, and the mRNA expression of p28, EBI3, WSX-1, and GP130 was detected by quantitative real-time PCR (qPCR). (**E**) The mouse bone marrow dendritic cells (BMDCs) were induced from naïve C57BL/6 mice, grouped as noninfected BMDCs (white), LPS-stimulated BMDCs (light grey), or Cm-infected BMDCs (deep grey) as described in the methods, stained with anti-WSX-1, and analyzed by flow cytometry. Data are shown as means ± SD, representing one of three independent experiments (n = 2–4/group/experiments). Statistical significances of differences are determined by one-way ANOVA. * *p* < 0.05, ** *p* < 0.01, *** *p* < 0.001, **** *p* < 0.0001. Abbreviations: DCs, dendritic cells; WSX-1, interleukin 27 receptor subunit alpha (IL-27Rα); *C. muridarum*/Cm, *Chlamydia muridarum*; IFUs, inclusion-forming units; p.i., post-infection; MFI, mean fluorescence intensity; EBI3, Epstein–Barr virus-induced gene 3; GP130, glycoprotein 130; LPS, lipopolysaccharide; BMDC, bone marrow dendritic cell.

**Figure 2 microorganisms-11-00604-f002:**
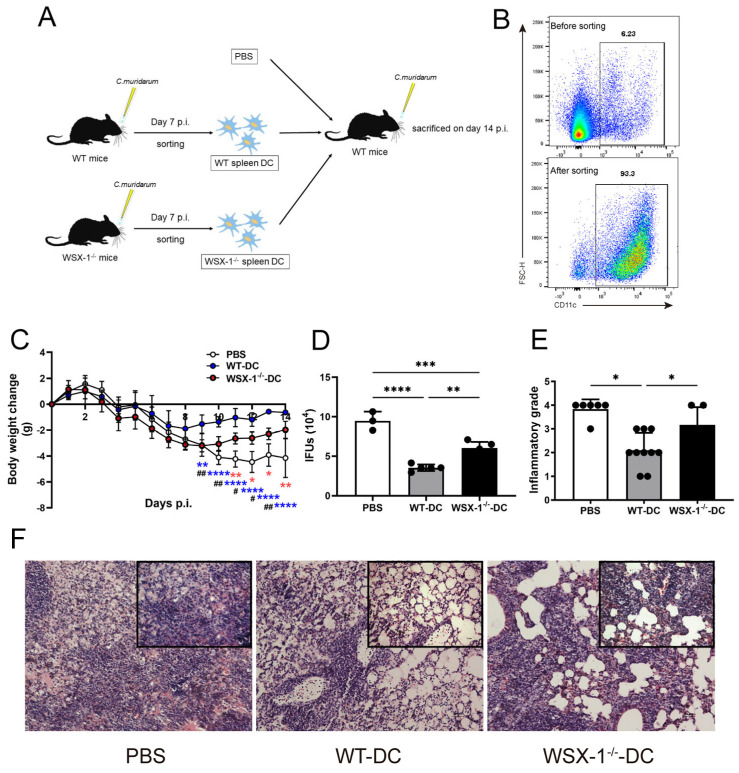
Disease progression of recipient mice after receiving DCs in *C. muridarum* lung infection. (**A**,**B**) WT mice and WSX-1-deficient (WSX-1^−/−^) mice were intranasally infected and sacrificed on day 7 p.i., spleen cells were isolated, and DCs were sorted with purity higher than 90% (**B**). Either 1 × 10^6^ WT-DCs or WSX-1^−/−^-DCs in 200 μL PBS were adoptively transferred to naïve C57BL/6 recipient mice by tail vein injection (the control group was given the same dose of PBS). Two hours after the adoptive transfer, the recipient mice were intranasally infected with 1 × 10^3^ IFUs *C. muridarum*. (**C**) The body weight changes were monitored daily and the mice were sacrificed on day 14 p.i. The red asterisks (*) represent statistical significances between the PBS and WSX-1^−/−^-DC recipients, blue asterisks (*) represent statistical significances between the PBS and WT-DC recipients, and the pounds (#) represent statistical significances between the WT-DC and WSX-1^−/−^-DC recipients. (**D**) The lung homogenates were prepared for determining chlamydia IFUs. (**E**,**F**) Lung sections were stained by H&E, the inflammatory grades were scored with the semi-quantitative pathological scoring method (**E**), and the representative pathological changes were captured under light microscopy [100X and 200X] (**F**). Data are shown as means ± SD, representing one (**C**,**D**) or two (**E**) of three independent experiments (n = 3–5/group/experiments). Statistical significances of differences are determined by two-way ANOVA (**C**) and one-way ANOVA (**C**–**E**). * *p* or # < 0.05, ** *p* or ## < 0.01, *** *p* < 0.001, **** *p* < 0.0001. Abbreviations: *C. muridarum*, *Chlamydia muridarum*; WSX-1^−/−^, IL-27Rα-deficient; p.i., post-infection; DC, dendritic cells; IFUs, inclusion-forming units; WT-DC, DC derived from wild-type mice; WSX-1^−/−^-DC, DC derived from WSX-1^−/−^ mice.

**Figure 3 microorganisms-11-00604-f003:**
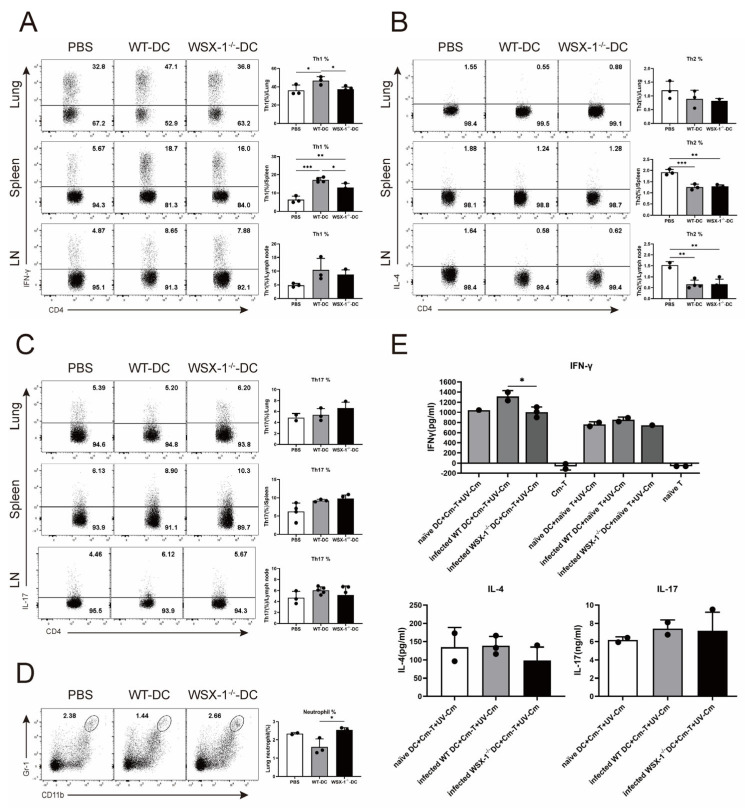
Th responses of recipient mice after receiving DCs in *C. muridarum* lung infection. The recipient mice were sacrificed on day 14 p.i. Lung, spleen, and lymph node single cells were prepared, and the Th responses were analyzed by flow cytometry and ELISA. (**A**–**D**) Representative flow cytometric images of IFN-γ-producing CD4^+^ T cells (Th1 cells, (**A**)), IL-4-producing CD4^+^ T cells (Th2 cells, (**B**)), IL-17-producing CD4^+^ T cells (Th17 cells, (**C**)), CD11b^+^ Ly-6G^+^ cells (neutrophils, (**D**)), and summaries of their percentages in the lung, spleen, and lymph node are shown. (**E**) Spleen DCs were sorted from naïve C57BL/6 mice, *C. muridarum*-infected C57BL/6 mice, and *C. muridarum*-infected WSX-1^-/-^ mice (naïve DC, infected WT-DC, and infected WSX-1^-/-^-DC) by magnetic microbeads. Spleen CD4^+^ T cells were sorted from naïve C57BL/6 mice and *C. muridarum*-infected mice (naïve T and Cm-T) by magnetic microbeads. The sorted DCs and CD4^+^ T cells were co-cultured for 48 h in the presence or absence of UV-sterilized *C. muridarum* (UV-Cm). IFN-γ, IL-4, and IL-17 in the culture supernatants were determined by ELISA. Data are shown as means ± SD, representing one of three independent experiments (n = 2–5/group/experiments). Statistical significances of differences are determined by one-way ANOVA. * *p* < 0.05, ** *p* < 0.01, *** *p* < 0.001. Abbreviations: *C. muridarum*/Cm, *Chlamydia muridarum*; WSX-1^-/−^, IL-27Rα-deficient; p.i., post-infection; LN, lymph node; UV-Cm, UV-sterilized *Chlamydia muridarum*; WT-DC, DC derived from wild-type mice; WSX-1^−/−^-DC, DC derived from WSX-1^−/−^ mice.

**Figure 4 microorganisms-11-00604-f004:**
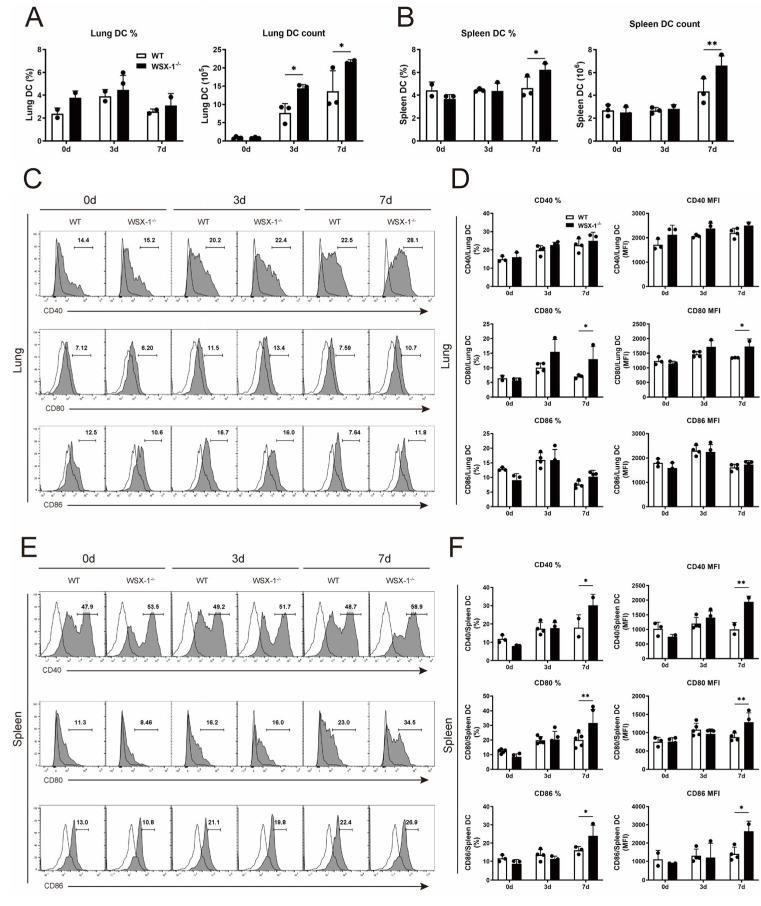
DCs accumulation and phenotypic maturation in WSX-1^−/−^ mice following *C. muridarum* respiratory infection. The lung and spleen single cells of WT and WSX-1^−/−^ mice on days 0, 3, and 7 p.i. were prepared. (**A**,**B**) The frequencies and total numbers of pulmonary DCs (**A**) and splenic DCs (**B**) were detected by flow cytometry. (**C**–**F**) The CD40, CD80, and CD86 expressions (shaded histogram) with fluorescence minus one (FMO) control (solid lines) on pulmonary DCs (**C**) and splenic DCs (**E**) were analyzed by flow cytometry. The percentages and MFI of positive cells at indicated times after infection were indicated (**D**,**F**). Data are shown as means ± SD, representing one of three independent experiments (n = 3–5/group/experiments). Statistical significances of differences are determined by two-way ANOVA. * *p* < 0.05, ** *p* < 0.01. Abbreviations: *C. muridarum*/Cm, *Chlamydia muridarum*; WSX-1^−/−^, IL-27Rα-deficient; p.i., post-infection; FMO, fluorescence minus one; MFI: mean fluorescence intensity.

**Figure 5 microorganisms-11-00604-f005:**
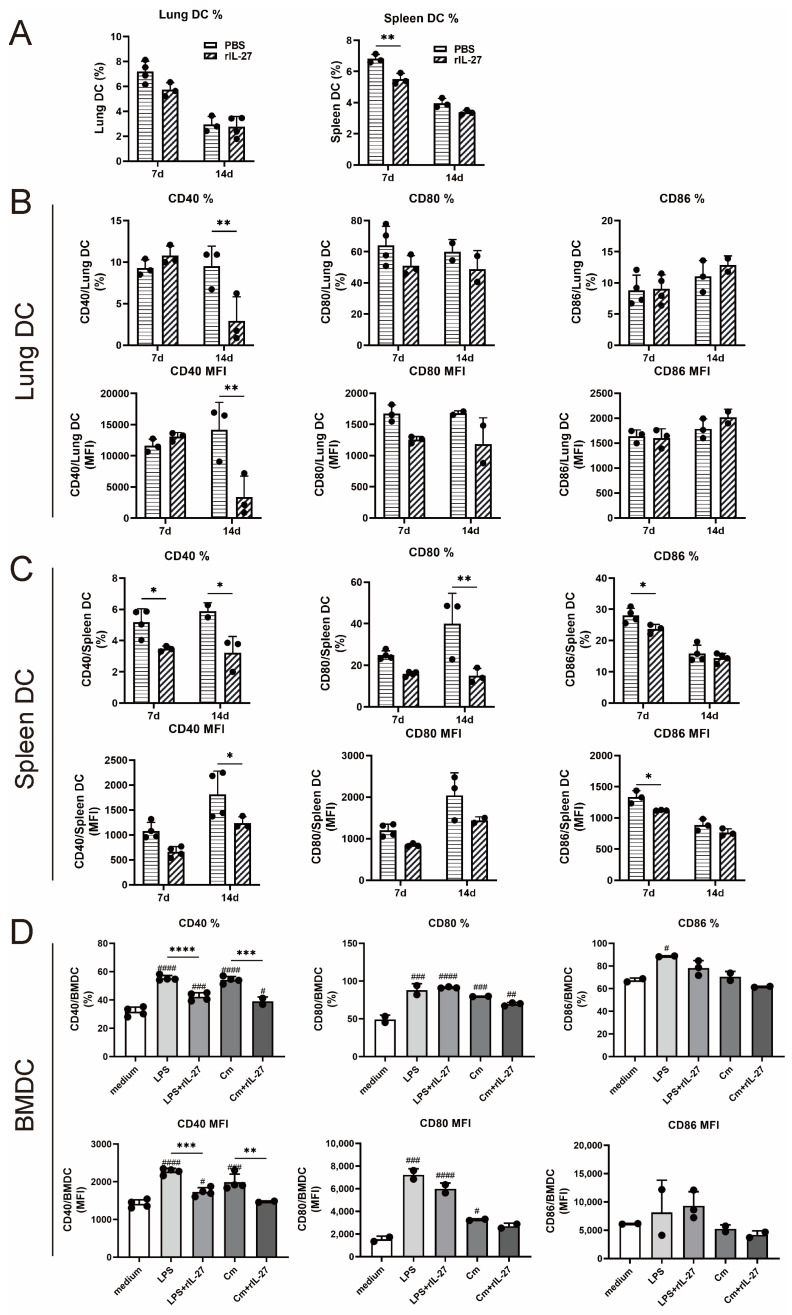
DCs accumulation and phenotypic maturation after rIL-27 administration against *C. muridarum* respiratory infection. (**A**–**C**) For recombinant murine IL-27 (rIL-27) administration, WT mice were inoculated intranasally with 0.2 μg rIL-27 in 30 μL PBS on the day before and days 0, 2, 4, and 6 after *C. muridarum* infection, and the control group was given 30 μL sterile PBS in the same schedule. The lung and spleen single cells on days 7 and 14 p.i. were prepared. (**A**) The percentages of pulmonary DCs and splenic DCs were detected by flow cytometry. Costimulatory molecules of DCs were further detected and altered expression of CD40, CD80, and CD86 on pulmonary DCs (**B**) and splenic DCs (**C**) was shown. (**D**) BMDCs were induced from naïve C57BL/6 mice and grouped as noninfected BMDCs, LPS-stimulated BMDCs, LPS-stimulated BMDCs with rIL-27 administration, Cm-infected BMDCs, and Cm-infected BMDCs with rIL-27 administration. Analyzed by flow cytometry, the percentages and MFI of CD40, CD80, and CD86 on BMDCs were indicated. Asterisks (*) represent statistical significances between two groups indicated by the lines, and the pounds (#) represent statistical significances compared to noninfected BMDCs. Data are shown as means ± SD, representing one of three independent experiments (n = 2–4/group/experiments). Statistical significances of differences are determined by two-way ANOVA (**A**–**C**) and one-way ANOVA (**D**). * or # *p* < 0.05, ** or ## *p* < 0.01, *** or ### *p* < 0.001, **** or #### *p* < 0.0001. Abbreviations: *C. muridarum*/Cm, *Chlamydia muridarum*; rIL-27, recombinant IL-27; p.i., post-infection; MFI: mean fluorescence intensity; BMDC, bone marrow dendritic cell.

**Figure 6 microorganisms-11-00604-f006:**
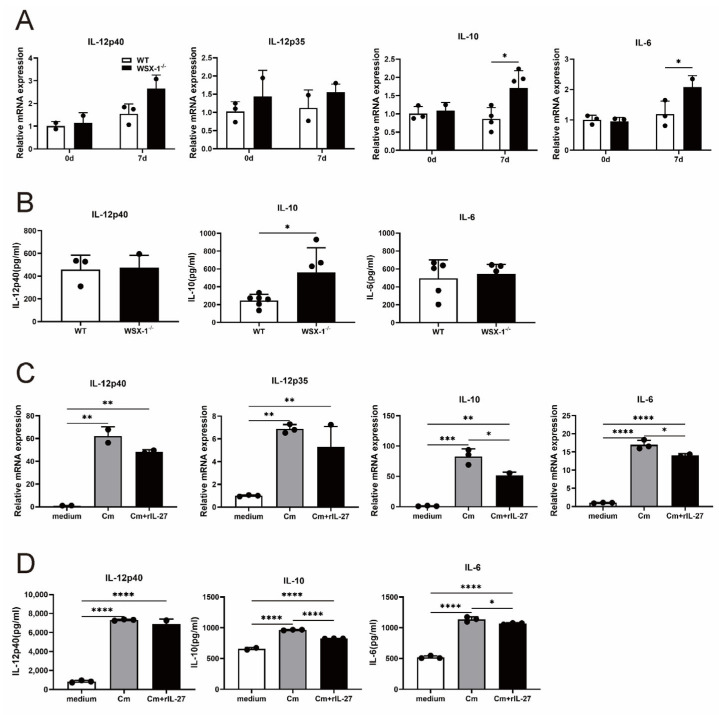
DCs cytokines production after WSX-1 deficiency or rIL-27 administration against *C. muridarum* infection. (**A**,**B**) The spleen DCs of WT and WSX-1^−/−^ mice on days 0 and 7 p.i. were sorted as described in the methods. The total RNA of sorted DCs was prepared for qPCR, and the mRNA expressions of cytokines IL-12p40, IL-12p35, IL-10, and IL-6 were shown (**A**). The culture supernatants of sorted DCs were prepared for ELISA, and IL-12p40, IL-10, and IL-6 productions in the culture supernatant were determined on day 7 p.i. (**B**). BMDCs were induced from naïve C57BL/6 mice, grouped as noninfected control BMDCs (white), Cm-infected BMDCs (grey), and Cm-infected BMDCs with rIL-27 administration (black), analyzed by qPCR (**C**) and ELISA (**D**) for cytokine production. Data are shown as means ± SD, representing one of three independent experiments (n = 3–6 /group/experiments). Statistical significances of differences are determined by two-way ANOVA (**A**), unpaired Student’s *t*-test (**B**), and one-way ANOVA (**C**,**D**). * *p* < 0.05, ** *p* < 0.01, *** *p* < 0.001, **** *p* < 0.0001. Abbreviations: *C. muridarum*/Cm, *Chlamydia muridarum*; WSX-1^−/−^, IL-27Rα-deficient; p.i., post-infection; BMDC, bone marrow dendritic cell; rIL-27, recombinant IL-27.

## Data Availability

The raw data used to support the findings of this study are available from the corresponding author upon request.

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
