# Peer review of "IL-27 Signaling Promotes Th1 Response by Downregulating IL-10 Production in DCs during Chlamydial Respiratory Infection"

_microorganisms, 2023, doi:10.3390/microorganisms11030604_

Round 1

Reviewer 1 Report

The work written by Zeng and co-workers is very interestin and well organized. However some changed are neeed before pubblication.

Introduction.

1-Line 60-62 are not clear. The authors should clearly explain the caracteristics of cytokines belonging IL12 super-family.

2-Line 84-87. IL10 is very important also for other functions and involvment in response to infective and non-infective stressors. Not only in TH1 response.

Discussion

3-line 439-441. If IL27 decrese IL10 Not only modulate TH1 response. This should be discuted.

4-Why didn’t the authors evaluate the effect on macrophages?

Reviewer 2 Report

The paper of Jiajia Zengal to be accepted:

The research is solid and well conducted. You can publish.

Author Response

Dear reviewer,

Many thanks to you for the detailed review of our manuscript. 

We feel grateful to receiving your approval for publishing.

Sincerely,

Hong Bai

Reviewer 3 Report

The presented study is a very complex work, shedding additional light on the functioning of the immune system during the development of chlamydial infection. The authors convincingly showed that IL-27/IL-27R mediates protection against chlamydial invasion by promoting a protective Th1 response. The work was done at a serious experimental level, including work with microorganisms and eukaryotic cells. All experimental data are statistically significant. The reviewer is not a very specialist in the field of immunology, so it is difficult for him to judge the intricacies of the work. However, all the presented visual materials, description of methods and conclusions look, at first glance, adequate. The authors of the manuscript have done a lot of work, which, in the opinion of the reviewer, deserves the attention of the scientific community, and can be published without major changes. The results of the study may be useful in developing strategies for the immune fight against persistent microbiota, including intracellular pathogens, such as chlamydia or mycoplasmas.

Author Response

(The authors gave the same response as above.)
